# Investigating Recurrent Transformers with Dynamic Halt

## Abstract

In this paper, we study the empirical effects of two major approaches to augmenting Transformers with a recurrent mechanism: (1) the approach of incorporating a depth-wise recurrence similar to Universal Transformers; and (2) the approach of incorporating a chunk-wise temporal recurrence like Temporal Latent Bottleneck. Furthermore, we propose and investigate novel ways to extend and combine the above methods - for example, we propose a global mean-based dynamic halting mechanism for Universal Transformers and an augmentation of Temporal Latent Bottleneck with elements from Universal Transformer. We compare the models and probe their inductive biases in several diagnostic tasks, such as Long Range Arena (LRA), flip-flop language modeling, ListOps, and Logical Inference. The code is released in the supplementary.

## 1  Introduction

Transformer (Vaswani et al., 2017) is a highly successful general-purpose attention-based architecture that aims to eliminate any recurrence or convolution. Although Transformers generally perform better than Recurrent Neural Networks (RNNs) given enough data[1], several works have shown that RNNs can still outperform Transformers in specific contexts - especially in algorithmic and structure-sensitive tasks under out-of-distribution (OOD) settings (Tran et al., 2018; Han et al., 2021; Shen et al., 2019a; Bhattamishra et al., 2023; Liu et al., 2023a; Deletang et al., 2023). Theoretical reasons (Han et al., 2021; Hao et al., 2022; Merrill et al., 2022; Liu et al., 2023b; Merrill & Sabharwal, 2023; Feng et al., 2023) are also suggested for such limitations of Transformers. Overall, these factors naturally raise the question of whether we can get "the best of both worlds" by combining recurrence and Transformers in the same model.

As a motivating example, consider a task like ListOps (Nangia & Bowman, 2018) where we have to solve a nested list of mathematical operations such as: `MAX(1,3,SUM(4,5,MIN(9,7)),4))`. Transformers utilize a self-attention mechanism where each position in a given sequence attends to all positions in the sequence. However, such an unconstrained all-to-all attention-based interaction is not immediately sufficient for a task like ListOps exemplified above. In the example, for instance, `5` cannot be summed up with `MIN(9,7)` immediately because the model has to wait until the MIN operation is applied. As such, the Transformer would need more layers to prepare intermediate computations (such as the output of `MIN(9,7)`) before other outer operations can be applied and the final result is computed. Each position having access to all positions via self-attention is not enough to bypass this requirement. However, the number of sequential operations required to get the final result can vary from example to example whereas standard Transformers are typically bound by a fixed number of layers that does not depend on the input. In contrast, the layer depth of RNNs can extend adaptively based on sequence length. To solve this task, a model has to process the sequence sequentially because the operations in the outer lists can only be executed once the inner list operations are executed. Keeping that in mind, we can list a few potentially necessary *conditions* or requirements to fulfill for a task like this (and other tasks like program synthesis, program execution, mathematical reasoning, algorithmic reasoning, etc.)[2]:

- **C1:** Ability to operate in arbitrary orders (for example, the innermost operations may occur in any position in the input).

---

[1]Newer RNNs like RWKV (Peng et al., 2023) or Mamba (Gu & Dao, 2024) could be potential exceptions.
[2]A similar list of requirements was also motivated in Csordás et al. (2022).

- **C2:** Being equipped with a gating mechanism to keep some hidden states unchanged if needed (for example, outermost values may need to remain unchanged for a while to wait for inner list operations to be completed).

- **C3:** Having an adaptive number of layers (according to the input) such that the number of layers can potentially increase indefinitely with higher demand (for example, when a higher depth of reasoning is required according to sequence length as longer sequences are more complex).

Transformers (Vaswani et al., 2017) with their fixed layers (failing C3) and lacking a gating mechanism to control information flow (failing C2) tend to struggle in tasks like ListOps (Tran et al., 2018; Shen et al., 2019a; Tay et al., 2021; Csordás et al., 2022) especially in out-of-distribution contexts - for example, when evaluated on data of higher sequence length compared to the length of training samples. Zhou et al. (2023) show that standard Transformers are most suited for length generalization when the problem can be solved by a short RASP-L program, failing otherwise. According to Merrill & Sabharwal (2023), log-precision non-recurrent Transformers fall under complexity class logspace-uniform $TC_0$.

Given these issues, we consider two broad directions of including recurrence in Transformers to tackle these limitations—one via adding depth-wise recurrence (§4.1, §5.1) and another via adding chunk-wise recurrence (§4.2, §5.2).

**1. Depth-wise Recurrence:** One way to introduce recurrence is to *repeat the same Transformer block over all tokens.* Such a strategy is employed by Universal Transformer (UT) (Dehghani et al., 2019). In the complete implementation, this is accompanied by a dynamic halting mechanism (Graves, 2016) that adaptively decides when to halt based on the input complexity. We consider Universal Transformer with a modern variant of its halting function (Tan et al., 2023) as the representative model to study for this direction (baseline UT).

**2. Chunk-wise Recurrence:** Another way to introduce recurrence is to *incorporate temporal recurrence just like RNNs do* except with a stack of Transformer blocks as the recurrent cell, using the past hidden states or some compressed representation as the internal RNN-like memory (Fan et al., 2020). Typically, this temporal recurrence is done at the level of chunks (subsequences) rather than single tokens to exploit the parallelism of Transformers (Didolkar et al., 2022; Ju et al., 2022; Hutchins et al., 2022; Bulatov et al., 2022). In other words, these models can recurrently process one whole chunk at a time rather than single tokens during training. We consider Temporal Latent Bottleneck (TLB) (Didolkar et al., 2022) as the representative model to study for this direction (baseline TLB).

Chunk-wise recurrence can also be backed by some practical engineering-related motivations. For example, Transformer XL (Dai et al., 2019) enables a recurrent cache mechanism to compress past information into a fixed number of memory slots for efficient language modeling. This bypasses the need for expensive attention to an indefinitely growing history of tokens. Other methods have extended this strategy with gradient propagation through the memory slots for long-range language modeling (Hutchins et al., 2022; Ju et al., 2022; Bulatov et al., 2022; Hwang et al., 2024). Some recent works have suggested that recurrent augmentation provides better memory for Transformers (Kuratov et al., 2024; Bulatov et al., 2024).

**Contributions:** Our main aim in this paper is to investigate and compare the potential and trade-offs of two different ways to incorporate recurrence for Transformers in algorithmic tasks that generally challenge standard Transformers. In doing so, we also propose and investigate some reasonable extensions and variations for earlier models. For instance, We propose Gated Universal Transformer (GUT) that uses a transition-aware global dynamic halting mechanism for Universal Transformer and integrates it with a gating mechanism and transition-dynamics-aware halting function. We also propose Gated Universal Transformer Latent Bottleneck (GUTLB) as a combination of GUT (which is a Depth-wise Recurrent Transformer) with TLB (which is a Chunk-wise Recurrent Transformer). We empirically investigate the advantages and disadvantages of the two forms of recurrences and our modifications in several diagnostic tasks such as Long Range Arena (LRA), flip-flop language modeling, ListOps, and Logical Inference.

We find that GUT tends to perform better than baseline UT in most of the algorithmic tasks tested here, but so far, we have not found much advantage for using GUTLB over TLB. We also show that chunk-

wise recurrence with its fixed recurrent attention window tends to show more robustness than depth-wise recurrence in length generalization and flip-flop language modeling but worse IID performance in certain structure-sensitive tasks like ListOps and logical inference. Note that our purpose is not to chase state-of-the-art results with our proposal but to investigate where they stand empirically.

## 2 Related Works

**Dynamic Halting:** Schmidhuber (2012) initially proposed the idea of using a halting neuron for self-delimiting neural programs. The work on Adaptive Computation Mechanism (ACT) by Graves (2016) laid the foundation for a lot of modern dynamic halting approaches. ACT was adapted into the original Universal Transformer (Dehghani et al., 2019) and later modified under probabilistic principles for PonderNet (Banino et al., 2021). Recently, Tan et al. (2023) adapted the ideas similar to Banino et al. (2021) and integrated them with a sparse Mixture of Expert mechanisms in their Sparse Universal Transformer. Our baseline halting mechanism is based on the halting algorithm used in Tan et al. (2023). In another direction, Deep Equilibrium Networks (DEQ) (Bai et al., 2019) implements UT in an implicit layer framework, which can be interpreted as having an implicit dynamic global-halting mechanism that stops the model when the hidden states converge. Xue et al. (2023) explored another alternative direction to standard dynamic halting where they focus on dynamically increasing the input sequence using additional representations generated by a prior RNN representing intermediate computational states to adapt to input complexity. Several works use some form of dynamic halting mechanism for early exit and accelerated inference (Han et al., 2022; Tan & Sim, 2016; Elbayad et al., 2020; Hou et al., 2020; Zhou et al., 2020; Xin et al., 2021; Schuster et al., 2021; 2022; Ainslie et al., 2023; Ji et al., 2023).

**Universal Transformer:** Universal Transformer (UT) strives to be Turing-complete like earlier works such as Neural Turning Machines (Graves et al., 2014) or Neural GPU (Kaiser & Sutskever, 2016). Several works have attempted to extend UT - some of which we already discussed above (Bai et al., 2019; Banino et al., 2021; Tan et al., 2023). ALBERT (Lan et al., 2020) scales UT with BERT-like training but forgoes the dynamic halting - although certain extensions of it do incorporate it (Balagansky & Gavrilov, 2022). Neural Data Router (Csordás et al., 2022) extends UT with geometric attention and gating. It gets strong results in several algorithmic tasks; however, it also forgoes dynamic halting, relying mainly on a gating mechanism to implicitly halt as needed while making it run up to some user-set upper bound number of layers.

**Gating:** Besides the work of Csordás et al. (2022), gating was also used in Transformers in an earlier work (Chai et al., 2020). Modern attention methods also integrate it with certain forms of gating (Shazeer, 2020; Hua et al., 2022; Qin et al., 2023a; Yang et al., 2023) inspired by the Gated Linear Unit (GLU) activation function (Dauphin et al., 2017).

## 3 Preliminaries

Here, we describe the building blocks of our models.

**Transformer:** In this paper, we focus on Transformer-based encoding models (Vaswani et al., 2017). Our Transformer block typically constitutes an attention layer followed by a feed-forward network. Given a sequence $H_l = (h_l^{(0)}, h_l^{(1)}, \ldots, h_l^{(n-1)}) \in \mathbb{R}^{n \times d}$ ($n$ being the sequence length and $d$ being the hidden state size) from layer $l$, we can formalize the layers as:

$$A_{l+1} = \text{Attention}(Q = H_l, K = H_l, V = H_l) \tag{1}$$

$$H_{l+1} = \text{FeedForward}(A_{l+1}) \tag{2}$$

The typical implementation of the Attention$(Q, K, V)$ layer can be formalized as:

$$Attention(Q, K, V) = \text{MHA}(LN(Q), LN(K), LN(V)) + Q \tag{3}$$

Here, $LN$ represents layer normalization (Ba et al., 2016) and $\text{MHA} : \mathbb{R}^{n \times d} \times \mathbb{R}^{n \times d} \times \mathbb{R}^{n \times d} \to \mathbb{R}^{n \times d}$ is the standard multiheaded attention (Vaswani et al., 2017). Eq. 3 shows pre-normalization with residual addition.

---

**Algorithm 1** Baseline Halting mechanism at a given timestep $t$

---

$\hat{\alpha}_{-1}^{(t)} = 0$
**for** $l = 1$ **to** $L$ **do**
  **if** $\sum_{l'=1}^{l-1} \alpha_{l'}^{(t)} < \alpha_{\text{thresh}}$ **then**
    $A_l = \text{Attention}(\underbrace{H_{l-1}}_{Q}, \underbrace{S_{l-1}}_{K}, \underbrace{S_{l-1}}_{V})$ # here, $S_{l-1} = (s_{l-1}^{(0)}, s_{l-1}^{(1)}, \dots, s_{l-1}^{(n-1)})$ and $H_{l-1} = (h_{l-1}^{(0)}, h_{l-1}^{(1)}, \dots, h_{l-1}^{(n-1)})$
    $H_l = \text{FeedForward}(A_l)$
    $\hat{\alpha}_{l-1}^{(t)} = \text{halt}(h_{l-1}^{(t)})$ # token-level halt
    $\alpha_{l-1}^{(t)} = \hat{\alpha}_{l-1}^{(t)} \prod_{l'=-1}^{l-2} (1 - \hat{\alpha}_{l'}^{(t)})$
    $s_l^{(t)} = \left(1 - \sum_{l'=0}^{l-1} \alpha_{l'}^{(t)}\right) \cdot h_l^{(t)} + \left(\sum_{l'=0}^{l-1} \alpha_{l'}^{(t)} \cdot h_{l'}^{(t)}\right)$
  **else**
    $h_l^{(t)} = h_{l-1}^{(t)}$
    $s_l^{(t)} = s_{l-1}^{(t)}$
  **end if**
**end for**

---

The FeedForward($A_l$) layer at any layer $l$ can be formalized as:

$$H_l = (\text{GeLU}(LN(A_l)W_1 + b_1)W_2 + b_2) + A_l \tag{4}$$

Here, $W_1 \in \mathbb{R}^{d \times d_{ff}}, W_2 \in \mathbb{R}^{d_{ff} \times d}, b_1 \in \mathbb{R}^{d_{ff}}$, and $b_2 \in \mathbb{R}^d$. $LN$ is again layer normalization. $d_{ff}$ is the size of the intermediate hidden states in the FeedForward function. GeLU is an activation function (Hendrycks & Gimpel, 2016). Eq. 4 shows a two-layered perceptron network with a pre-normalization strategy and a residual addition.

## 4 Prior Works

Here, we discuss prior works implementing depth-wise and chunk-wise recurrences in Transformers that we use as baselines.

### 4.1 Depth-wise Recurrence: Universal Transformer

One approach to tackling condition C3 above via depth-wise recurrence is to use Universal Transformers (UT) (Dehghani et al., 2019) instead of the standard Transformer. Briefly, UT repeatedly applies the same Transformer block in every layer while a halting mechanism decides when to terminate the repetition based on the hidden state. Reusing the same block (thus, sharing the parameters) can allow better function reuse (as similar functional operations can be required in different layers) and allows the model to adapt flexibly to task complexity without any hard preset upper bound. In principle, it is also possible to simulate any vanilla Transformer with unshared parameters with parameter-shared Transformers given enough hidden state size as shown in Bai et al. (2019). However, in practice, we still need to set some upperbound for the number of layers because we do not want our models to add layers indefinitely. In the case of vanilla Transformers, we have to also worry about the number of parameters when setting the upperbound because more layers imply more parameters. In case of Universal Transformers, we do not need to worry about parameters to set the upperbound because the parameters are shared across all the layers. Nevertheless, we still need to worry about latency and computational resource when increasing layers. The standard strategy, given a upperbound, is to run through all the layers until the upperbound is reached. This strategy however is not ideal. In practice, some inputs may require very little layers whereas some other inputs may require the maximum allowed number of layers Graves (2016). As such, the standard strategy results in a dilemma. Either, we can lower the upperbound to save compute and latency at the cost of ability to handle some of the more complex inputs or we have to keep a high upperbound to maintain coverage of more complex inputs but at the severe cost of latency and unnecessary compute for simpler inputs. Dynamic halting is a

mechanism that provides a way out of this dilemma. The dynamic halting mechanism is designed to learn when to halt (i.e. learn in which layer to terminate) based on the given input. In the ideal setup, we can thus have a reasonably high upperbound to get coverage of complex examples, but at the same time, using the dynamic halting mechanism a model can save on compute and latency by halting early in case of simpler examples.

**Dynamic Halting:** The baseline algorithm that we use for UT is shown in Algorithm 1. The algorithm is mainly adapted from Tan et al. (2023) with minor presentational modifications for consistency with our formalism. The standard halting mechanism used in UT is token-level, i.e., different token representations $(h_l^{(t)})$ at different time steps $(t)$ can halt at different layers $l$. After halting, the token representation will no longer be updated. We implement the $\text{halt}(h_{l-1}^{(t)})$ function (as used in Algorithm 1) as follows:

$$\text{halt}(h_{l-1}^{(t)}) = \sigma_2(\sigma_1(h_{l-1}^{(t)}W_{h1} + b_{h1})W_{h2} + b_{h2}) \tag{5}$$

Here, $h_{l-1}^{(t)} \in \mathbb{R}^d$, $W_{h1} \in \mathbb{R}^{d \times d}, W_{h2} \in \mathbb{R}^{d \times 1}, b_{h1} \in \mathbb{R}^d$, and $b_{h2} \in \mathbb{R}$. $\sigma_1$ is GeLU activation and $\sigma_2$ is sigmoid activation. The algorithm implements a soft probabilistic halting procedure to compute halting probability per layer. Here, $\hat{\alpha}_{l-1}^{(t)}$ represents the probability of halting at layer $l-1$ and position $t$ conditioned on the event that the model has not already halted at position $t$. Thus, the unconditional halting probability at layer $l-1$ and position $t$ $(\alpha_{l-1}^{(t)})$ can be calculated from probabilistic principles as done in Algorithm 1:

$$\alpha_{l-1}^{(t)} = \hat{\alpha}_{l-1}^{(t)} \prod_{l'=-1}^{l-2} (1 - \hat{\alpha}_{l'}^{(t)}) \tag{6}$$

The final output $(S_L = (s_l^{(0)}, s_l^{(1)}, \ldots, s_l^{(n-1)})$ in Algorithm 1) is a marginalization of all layer outputs (produced before the unconditional halting probability exceeds a threshold $\alpha_{thresh}$) weighted based on their halting probability which represents the probability of the corresponding representation being the *final* representation. In other words, we softly select the final output per position. As shown in Algorithm 1, this is mathematically represented as:

$$s_l^{(t)} = \left(1 - \sum_{l'=0}^{l-1} \alpha_{l'}^{(t)}\right) \cdot h_l^{(t)} + \left(\sum_{l'=0}^{l-1} \alpha_{l'}^{(t)} \cdot h_{l'}^{(t)}\right) \tag{7}$$

We can early halt (before going through all the layers until the upperbound) if the unconditional halting probability (which is bound to monotonically increase with layer number) at a layer exceeds a hyperparameter threshold $(\alpha_{thresh})$ because we do not use any representation after that layer for the final output. This way, we can save compute compared to running the model for the maximum number of layers. We allow early stopping both during training and inference.

The baseline algorithm captures the ideas from Banino et al. (2021), which re-formalizes an earlier halting mechanism (Graves, 2016; Dehghani et al., 2019) in a probabilistically principled manner.

**Auxiliary Loss:** Similar to Tan et al. (2023), we also set an auxiliary cost (loss) which would discourage halting too late:

$$\mathcal{L}_{\text{ACT}} = \frac{1}{T} \sum_{t=1}^{T} \sum_{l=1}^{L} \alpha_l^{(t)} \cdot l. \tag{8}$$

Here $t$ represents the position in a sequence, $T$ is the sequence length, $L$ is the total number of layers, $l$ is the layer number, $\alpha_l^{(t)}$ is the probability of halting for position $t$ at layer $l$. Late halting is equivalent to putting more probability mass (high $\alpha_l^{(t)}$ values) to later layers (higher $l$). Thus, late halting corresponds to high $\mathcal{L}_{\text{ACT}}$ - making it work as a penalty function to discourage late halting. The model is fully terminated when the cumulative halting probability of every timestep passes a certain threshold $\alpha_{\text{thresh}}$. Note the weight given to the objective $\mathcal{L}_{\text{ACT}}$ typically requires difficult tuning.

---

**Algorithm 2** Transformer Latent Bottleneck

---

GIVEN: $M^0$ #initial memory state
GIVEN: $g$ # g = chunk size (hyperparameter)
$k = \lceil len(H_0)/g \rceil$
**chunks** = torch.chunk$(H, k)$ #k = number of chunks
**for** $t, C_0$ **in** enumerate(**chunks**) **do**
   #Initially $t$=0
   # Chunk Processing
   **for** $l = 1$ **to** $L$ **do**
      $A_l = \text{Attention1}_l(\underbrace{C_{l-1}}_{Q}, \underbrace{C_{l-1}}_{K}, \underbrace{C_{l-1}}_{V})$
      # Integrating past memory
      $A_l = \text{Attention2}_l(\underbrace{A_l}_{Q}, \underbrace{M^t}_{K}, \underbrace{M^t}_{V})$
      $C_l = \text{FeedForward}_l(A_l)$
   **end for**
   # Recurrent memory update
   $M_0^t = M^t$
   **for** $l = 1$ **to** $U$ **do**
      $A_l = \text{Attention3}_l(\underbrace{M_{l-1}^t}_{Q}, \underbrace{C_L}_{K}, \underbrace{C_L}_{V})$
      $M_l^t = \text{FeedForward}_l(A_l)$
   **end for**
   $M^{t+1} = M_U^t$
**end for**

---

## 4.2 Chunk-wise Recurrence: Temporal Latent Bottleneck

Another approach to introducing recurrence and dynamic depth to Transformers is to make Transformers temporally recurrent (Fan et al., 2020), similar to standard Recurrent Neural Networks (RNNs). However, processing one token at a time recurrently with a non-linear Transformer block can be exorbitantly slow during training. Thus, to keep some recurrence for its benefits and also gain practical benefits of parallelism from Transformer - a popular strategy is to first divide the given sequence into temporal chunks (or subsequences) and then recurrently process one whole chunk at a time with a Transformer-based recurrent cell. This results in a chunk-wise recurrent processing. Many approaches explore a similar direction (Hutchins et al., 2022; Ju et al., 2022; Bulatov et al., 2022; Rae et al., 2020; Wu et al., 2022b; Lei et al., 2020). Here, we consider a recent method - Temporal Latent Bottleneck (TLB) (Didolkar et al., 2022) as a representative[3] given that its performance has been well demonstrated in a wide variety of tasks.

The basic algorithm used in TLB is detailed in Algorithm 2. Attention1, Attention2, and Attention3 in the algorithm are the same attention functions (as MHA described before) structurally, just named differently to denote that they have different parameters. $M^t \in \mathbb{R}^{k \times d}$ is a sequence of hidden states representing an RNN-like internal memory compressing past information into some fixed memory slots ($k$ represents the number of slots). We slightly modified the original algorithm to make it more comparable to the proposed extension that we discuss later (§5.2). Particularly, in the original method (Didolkar et al., 2022), the Attention2 is not used in every layer - but only periodically, and separate feedforward functions are used after each attention function (unlike in our case). Our implementation baseline simplifies the design.

TLB is interleaved with both self-attention and cross-attention layers. TLB uses self-attention to refine the hidden states in a given chunk. TLB uses cross-attention to integrate information from the past chunks with the hidden states representing the current chunk by having the current chunk states attend to the recurrently encoded memory states representing the past chunks. At the end of processing a given chunk, TLB updates the recurrent memory states with information from the current chunk by having the memory states encoding

---

[3]We also briefly explored Recurrent Memory Transformer (Bulatov et al., 2022) in LRA in preliminary experiments and found it lacking compared to TLB.

the past chunks attend to the last hidden states created for the current chunk through the earlier interleaved self-attention and cross-attention layers.

### 4.3 Limitations

**Depth-wise Recurrence:** In the standard UT, a token-level halting mechanism is implemented; that is, the halting probability is tracked separately for each time step or position. The model is only fully terminated when all the positions are individually halted (or some upper bound is reached). Moreover, once halted, they are permanently halted - in other words, the representation at the halted position becomes read-only for all subsequent layers.

However, in practice, for hierarchical tasks like mathematical reasoning or program synthesis, the model may need to only temporarily restrict updates at some positions (e.g., while waiting for computation in the inner lists to be completed in a task like ListOps). For example, given a sequence "4 x (5 + 6) - 7" (assuming a whitespace-based tokenization) states related to 4,x,-, and 7 has to wait until the first priority operation "(5+6)" is completed. While, in theory, Transformers may be able to implicitly learn a gating mechanism to preserve relevant information in some section of the hidden states across multiple layers, in practice having a more explicit mechanism can provide a better inductive bias in learning these desired capabilities. The benefit of an explicit gating mechanism for algorithmic tasks was empirically shown by Csordás et al. (2022).

Moreover, the token-level halt increases the number of halting decisions needed to determine the final halt, thus potentially increasing the chances of halting errors per sample (making some positions halt too early). In addition, the token-level halt can limit the writable memory (each position can also be considered as a memory slot with read-write capacities) by halting certain positions too early but without providing any computational saving in return because the model still has to wait for *all* the positions to halt to terminate fully.

Furthermore, typically, in UT, the scoring for halting at any layer $l$ is computed only using the hidden state at layer $l$ ($H_l$). Thus, it can lack foresight from future layers as to whether the next transition can be helpful or not in deciding whether to halt at the current time.

**Chunk-wise Recurrence:** To an extent, chunk-wise recurrent Transformers can adapt to input complexity dynamically - for example, the number of recurrent steps they execute depends on the number of chunks that exist, which depends on the given input sequence size. However, it lacks flexibility in the degree to which it adapts. For instance, some chunks may require more compute than others. One of the original motivations for an early dynamic halting mechanism (Adaptive Computation Time or ACT for short) (Graves, 2016) was to adapt compute based on the significance of the token being processed in the given time step. A similar motivation applies to chunk-wise recurrence, too. Different chunks may require different levels of compute resources based on the nature of the input chunk at the time step.

## 5 Proposed Extensions

In this section, we present our proposed extensions that address the aforementioned limitations.

### 5.1 Depth-wise Recurrence: Gated Universal Transformer

Given the aforementioned limitations for depth-wise recurrence, we propose Gated Universal Transformer (GUT). In GUT, we make the following modifications:

1. We use an explicit **gating mechanism** so that the model can learn to temporarily restrict updates to positions in a cleaner manner without permanently halting them. This also better addresses C2.

2. We introduce a **global halting mechanism** to decide halting based on a single global state representation.

3. We make the halting decision **transition-aware** by using the information from the immediate future layer.

---

**Algorithm 3** Proposed Global Halting mechanism at a given timestep $t$

---

$\hat{\alpha}_{-1}^{(t)} = 0$
**for** $l = 1$ **to** $L$ **do**
    **if** $\sum_{l'=1}^{l-1} \alpha_{l'} < \alpha_{\text{thresh}}$ **then**
        $A_l = \text{Attention}(\underbrace{H_{l-1}}_{Q}, \underbrace{H_{l-1}}_{K}, \underbrace{H_{l-1}}_{V})$ # here, $H_{l-1} = (h_{l-1}^{(0)}, h_{l-1}^{(1)}, \ldots, h_{l-1}^{(n-1)})$
        $H_l = \text{GatedFeedForward}(H_{l-1}, A_l)$ # gating
        $trans = [mean(H_{l-1}); mean(H_l)]$ # transition dynamics
        $\hat{\alpha}_{l-1} = \text{halt}(trans)$ # mean-based global halt
        $\alpha_{l-1} = \hat{\alpha}_{l-1} \prod_{l'=-1}^{l-2} (1 - \hat{\alpha}_{l'})$
        $S_l = \left(1 - \sum_{l'=0}^{l-1} \alpha_{l'}\right) \cdot H_l + \left(\sum_{l'=0}^{l-1} \alpha_{l'} \cdot H_{l'}\right)$
    **else**
        break
    **end if**
**end for**

---

We show the algorithm of GUT in Algorithm 3. The main changes in Algorithm 3 over Algorithm 1 are highlighted in blue. We discuss the modifications in more detail below.

**(1) Gating:** To implement gating, we replace the standard feed-forward network with a gated mechanism (GatedFeedForward in the algorithm). The gating mechanism that we use is similar to that in Csordás et al. (2022), but they lack any dynamic halting mechanism for early termination. The processing of $H_l = \text{GatedFeedForward}(H_{l-1}, A_l)$ can be broken down to the following equations:

$$H_{inter} = \text{FeedForward}(A_l) \tag{9}$$

$$G = \sigma_2(\sigma_1(LN(A_l)W_{g1} + b_{g1})W_{g2} + b_{g2}) \tag{10}$$

$$H_l = G \odot H_{inter} + (1 - G) \odot H_{l-1} \tag{11}$$

Here, $W_{g1} \in \mathbb{R}^{d \times d_{gff}}, W_{g2} \in \mathbb{R}^{d_{gff} \times d}, b_{g1} \in \mathbb{R}^{d_{gff}}$, and $b_{g2} \in \mathbb{R}^d$. $\sigma_1$ is GeLU activation and $\sigma_2$ is sigmoid activation. The FeedForward function is the same as before.

**(2) Global Halt:** In our global halt mechanism, the halting function determines the halting probability based on some pooling over the whole hidden sequence rather than the hidden state of a particular time step:

$$\hat{\alpha}_{l-1} = \text{halt}(pool(H_{l-1})) \tag{12}$$

We use mean-pooling in our specific implementation:

$$\hat{\alpha}_l = \text{halt}(mean(H_{l-1})) \tag{13}$$

Here, $\hat{\alpha}_{l-1}$ represents the probability of halting at layer $l - 1$ conditioned on the event the model has not already halted. Note that the current halting probability is no longer indexed with $t$ because it does not vary from position to position. Rather, we have a global halting probability for all positions in a specific layer.

Mean-pooling (mean: $\mathbb{R}^{n \times d} \rightarrow \mathbb{R}^d$) involves a mean operation over the temporal axis. Thus given $H_{l-1} = (h_{l-1}^{(0)}, h_{l-1}^{(1)}, \ldots, h_{l-1}^{(n-1)})$, where any $h_{l-1}^{(i)} \in \mathbb{R}^d$, the mean operation can be represented as:

$$mean(H_{l-1}) = \frac{1}{n} \sum_{i=0}^{n-1} h_{l-1}^{(i)} \tag{14}$$

Similar to before, the unconditional halting probability at layer $l-1$ ($\alpha_{l-1}$) can be calculated from $\hat{\alpha}_{l-1}$ using probabilistic principles (as shown in Algorithm 3):

$$\alpha_{l-1} = \hat{\alpha}_{l-1} \prod_{l'=-1}^{l-2} (1 - \hat{\alpha}_{l'}) \tag{15}$$

The final output $S_l$ is again the marginalization of all layer outputs that are produced before the threshold ($\alpha_{thresh}$) is exceeded based on the corresponding halting probabilities as shown in Algorithm 3:

$$S_l = \left(1 - \sum_{l'=0}^{l-1} \alpha_{l'}\right) \cdot H_l + \left(\sum_{l'=0}^{l-1} \alpha_{l'} \cdot H_{l'}\right) \tag{16}$$

Early halting works the same way as before. The auxiliary loss for discouraging late halting in Eq. 8 can also be simplified as $\mathcal{L}_{\text{ACT}} = \sum_{l=1}^{L} \alpha_l \cdot l$.

Besides this, when gating is present, it can also simulate token-level halting, if needed, by continuously zeroing out certain positions in every layer.

**(3) Transition-based halt:** Here, we seek to enrich the input to the halt function (to decide whether to halt at some layer $l-1$) by entering both the hidden state of layer $l-1$ and also the immediate future hidden state ($H_l$). Thus, the halting function is informed by the future hidden state to consider whether to halt now. Combined with mean-pooling-based global halting, transition-based scoring for the halting-probability can be represented as:

$$\hat{\alpha}_{l-1} = \text{halt}([mean(H_{l-1}); mean(H_l)]) \tag{17}$$

Here, [; ] represents concatenation. The shape of the first layer weight $W_{h1}$ in halt function is changed to $\mathbb{R}^{2d \times d}$ to account for the increased vector size. Balagansky & Gavrilov (2022) also used a form of transition-aware halting, but their approach lacks gating mechanism and global halting.

## 5.2 Chunk-wise Recurrence: Gated Universal Temporal Latent Bottleneck

Based on the limitation of chunk-wise recurrence that we discussed, we introduce a dynamic halting mechanism at every chunk. More precisely, we replace the transformer-based recurrent function in TLB with a GUT-based function. We call this combination the Gated Universal Transformer Latent Bottleneck (GUTLB). The algorithm for the combined model is shown in Algorithm 4. The main changes compared to TLB are highlighted in blue.

GUTLB can also be seen as a depth-wise recurrent Transformer-like UT where the attention is locally windowed, and the sequence length regulates the halting time. For instance, the longer the sequence, the more chunks there will be to process, and thus, there will be more recurrent layers of processing until complete termination, irrespective of how early the per chunk-level processing is halted. This also makes good sense as an inductive bias for halting because, intuitively, higher sequence length, in general, can indicate higher complexity and, thus, more computational need.

## 5.3 Depth-wise vs. Chunk-wise

Here, we discuss one salient distinction between standard depth-wise recurrence (UT, GUT) and chunk-wise recurrence (TLB, GUTLB) that motivates our comparison. For UT or GUT - all tokens are available for attention at every level. This may introduce noise through the dense softmax-based attention, making it harder to generalize to higher lengths or deal with sensitive tasks (where one token change can change the ground truth) (Liu et al., 2023a; Bhattamishra et al., 2023). Attention sparsifying and sharpening techniques have been used in consideration for making the attention mechanism more robust for these contexts (Liu et al., 2023a). But another alternative can be to bound the attention receptive field to a fixed chunk size by using chunk-wise recurrence (TLB, GUTLB). There is already some evidence that chunk-wise recurrent/recursive Transformers can better perform state tracking (Ju et al., 2022) in sensitive synthetic tasks, parity detection

---

**Algorithm 4** Gated Universal Transformer Latent Bottleneck

---

GIVEN: $M^0$ #initial memory state
GIVEN: $g$ # g = chunk size (hyperparameter)
$k = \lceil len(H_0)/g \rceil$
**chunks** = torch.chunk($H_0, k$) #k = number of chunks
$\hat{\alpha}^{(t)}_{-1} = 0$
**for** $t, C_0$ **in** enumerate(**chunks**) **do**
   #Initially $t=0$
   **for** $l = 1$ **to** $L$ **do**
     **if** $\sum_{l'=1}^{l-1} \alpha_{l'} < \alpha_{\text{thresh}}$ **then**
      $A_l = \text{Attention1}(\underbrace{C_{l-1}}_{Q}, \underbrace{C_{l-1}}_{K}, \underbrace{C_{l-1}}_{V})$
      $A_l = \text{Attention2}(\underbrace{A_l}_{Q}, \underbrace{M^t}_{K}, \underbrace{M^t}_{V})$
      $C_l = \text{GatedFeedForward}(H_{l-1}, A_l)$
      $trans = [mean(C_{l-1}); mean(C_l)]$
      $\hat{\alpha}_{l-1} = \text{halt}(trans)$  #mean-based global halt
      $\alpha_{l-1} = \hat{\alpha}_{l-1} \prod_{l'=-1}^{l-2} (1 - \hat{\alpha}_{l'})$
      $S_l = \left(1 - \sum_{l'=0}^{l-1} \alpha_{l'}\right) \cdot C_l + \left(\sum_{l'=0}^{l-1} \alpha_{l'} \cdot C_{l'}\right)$
     **else**
      break
     **end if**
   **end for**
   $M^t_0 = M^t$
   # Recurrent memory update
   **for** $l = 1$ **to** $U$ **do**
     $A_l = \text{Attention3}_l(\underbrace{M^t_{l-1}}_{Q}, \underbrace{S_L}_{K}, \underbrace{S_L}_{V})$
     $M^t_l = \text{FeedForward}_l(A_l)$
   **end for**
   $M^{t+1} = M^t_U$
**end for**

---

(Chi et al., 2023), and length generalization (Didolkar et al., 2022). We further explore the potential of chunk-wise recurrence here compared to depth-wise recurrence - mostly in contexts where they are not yet evaluated side-by-side.

# 6 Experiments and Results

## 6.1 Implementation details

For all models, we implement the multiheaded attention mechanism with FlashAttention2 (Dao, 2024). For positional encoding, we used xPos (Sun et al., 2023b). Other hyperparameter details are presented in Appendix A.

## 6.2 Choice of Models

Our main aim is to investigate the effects of incorporating different recurrent inductive biases into moderately standard Transformers. Based on that, we concentrate on mainly relevant models and baselines for our experiments. Thus, we ignore other unorthodox Transformers, recursive neural networks, state space models, and such, even though some of them may achieve state-of-the-art results in the datasets we explore. However, we discuss future works related to some of these models in §9.

| Model | near-IID | Length Gen. | | | Argument Gen. | | LRA |
|---|---|---|---|---|---|---|---|
| (Lengths) | $\leq 1000$ | 200-300 | 500-600 | 900-1000 | 100-1000 | 100-1000 | 2000 |
| (Arguments) | $\leq 5$ | $\leq 5$ | $\leq 5$ | $\leq 5$ | 10 | 15 | 10 |
| Transformer * | $57.4_{0.4}$ | — | — | — | — | — | — |
| UT * | $71.5_{7.8}$ | — | — | — | — | — | — |
| Our Implementations | | | | | | | |
| Transformer | $73_{0.79}$ | $15.6_{1.5}$ | $10.4_{0.71}$ | $9.6_{0.67}$ | $15.8_{1.1}$ | $10.3_{1.2}$ | $8.1_{0.75}$ |
| UT | $65.8_{7.3}$ | $14.1_{1.6}$ | $10.3_{0.99}$ | $10_{0.44}$ | $13.9_{0.59}$ | $10.7_{0.29}$ | $8.6_{0.81}$ |
| GUT | $\mathbf{83.2_{1.1}}$ | $17.6_{2.5}$ | $11.1_{0.96}$ | $10.5_{0.84}$ | $16.2_{2.1}$ | $14.2_{2.4}$ | $9.8_{1.1}$ |
| $-$Global Halt | $58.9_{0.89}$ | $11_{0.67}$ | $11.6_{0.3}$ | $9.8_{0.29}$ | $10.7_{0.66}$ | $10.7_{1.9}$ | $10.2_{1.3}$ |
| $-$Gate | $59.1_{2.7}$ | $15.2_{2.5}$ | $12.9_{1.9}$ | $11.2_{1.8}$ | $14.7_{1.8}$ | $12.7_{1.1}$ | $10.7_{1.2}$ |
| $-$Transition | $77.7_{5.5}$ | $13.8_{0.36}$ | $11.9_{0.84}$ | $9.7_{0.33}$ | $15.5_{1.7}$ | $11.9_{1.9}$ | $9.6_{0.82}$ |
| TLB | $58.3_{0.73}$ | $\mathbf{37.7_{1.6}}$ | $\mathbf{30.5_{1.1}}$ | $\mathbf{25.0_{2.0}}$ | $\mathbf{37.3_{0.75}}$ | $\mathbf{37.0_{1.1}}$ | $\mathbf{31.1_{1.3}}$ |
| GUTLB | $63.6_{3.5}$ | $\mathbf{35.8_{2.4}}$ | $\mathbf{28.4_{3.4}}$ | $21_{4.8}$ | $\mathbf{34.5_{2.8}}$ | $\mathbf{33.8_{3.9}}$ | $\mathbf{26.4_{5.3}}$ |

Table 1: Accuracy on ListOps. ListOps uses the original training set (Nangia & Bowman, 2018) (after removing sequences of length $> 100$ ) with the length generalization splits from Havrylov et al. (2019), the argument generalization splits from Ray Chowdhury & Caragea (2023a), and the ListOps-LRA test set from Tay et al. (2021). * represents that the results are taken from Shen et al. (2019a). Our models were trained on lengths $\leq 100$, depth $\leq 20$, and arguments $\leq 5$. For our models, we report the mean of 3 runs. We bold the best results (and any results that overlap with it in their standard deviation) in the first and second blocks separately. Subscript represents standard deviation. As an example, $90_{1.0} = 90 \pm 1.0$.

## 6.3 Task Motivations

We empirically investigate the models in tasks like ListOps Nangia & Bowman (2018), logical inference (Bowman et al., 2015), Flip-flop language modeling (Liu et al., 2023a), and Long Range Arena (LRA) (Tay et al., 2021). ListOps and logical inference are interesting to explore because Transformers have been shown to underperform in this task despite its simplicity (Shen et al., 2019a)[4]. This task can also be a diagnostic testbed for evaluating how well the dynamic halting function is being learned to adapt to the underlying tree depths of the tasks. Solving this task is not as easily hackable through shortcuts or spurious correlations - instead, the model needs to learn to halt at the right time. Flip-flop language modeling is an interesting diagnostic task where vanilla Transformers tend to struggle (Liu et al., 2023a) - thus making it a useful testbed for exploring the benefits of recurrent inductive bias in standard Transformers. LRA is a useful testbed for evaluating the models on a few moderately realistic tasks and also test long-range processing capabilities.

## 6.4 ListOps

**Description:** ListOps or List Operations is an arithmetic task for solving nested list of mathematical operations as exemplified in §1. ListOps represents the ListOps split set up similarly to Ray Chowdhury & Caragea (2023a). The training split is the original ListOps (Nangia & Bowman, 2018) with any sample with length $> 100$ size being filtered (except for the case of results copied from (Shen et al., 2019a)). The testing splits consist of several splits of higher lengths, including the original testing split ('near-IID') of mixed lengths (mainly similar size lengths to the training split), some split with a higher number of maximum arguments for each list operation, and the testing split from Long Range Arena (LRA) (Tay et al., 2021) which is both longer and has more arguments.

---

[4]Neural Data Router (Csordás et al., 2022) is a Transfromer-based model that shows decent performance in ListOps, but they train with 10× more data and short sequence lengths ($\sim 50$). They still struggle with generalization to higher sequence lengths or argument generalization (Ray Chowdhury & Caragea, 2023a).

| Model | Number of Operations | | | | | |
|---|---|---|---|---|---|---|
| | 7 | 8 | 9 | 10 | 11 | 12 |
| Transformer* | 51 | 52 | 51 | 51 | 51 | 48 |
| UT* | 51 | 52 | 51 | 51 | 51 | 48 |
| SUT† | **98** | **97** | **94** | **90** | **88** | **81** |
| Our Implementations | | | | | | |
| Transformer | $90.88_{0.41}$ | $81.26_{1.80}$ | $\mathbf{72.56_{2.4}}$ | $\mathbf{64.71_{4.54}}$ | $\mathbf{58.45_{3.1}}$ | $\mathbf{53.88_{4.4}}$ |
| UT | $76.65_{1.24}$ | $65.47_1$ | $57.4_{1.73}$ | $51.02_{1.42}$ | $50.08_{1.95}$ | $46.62_{0.11}$ |
| GUT | $\mathbf{96.36_{0.23}}$ | $\mathbf{84.84_{1.42}}$ | $\mathbf{74.80_{1.65}}$ | $\mathbf{66.81_{2.43}}$ | $58.99_{3.69}$ | $51.97_{2.83}$ |
| − Global Halt | $80.30_{8.06}$ | $67.55_{8.33}$ | $58.69_{8.04}$ | $54.18_{6.43}$ | $50.35_{5.81}$ | $45.92_{4.6}$ |
| − Gate | $93.12_{1.30}$ | $82.65_{1.55}$ | $72.08_{3.12}$ | $65.63_{2.48}$ | $\mathbf{58.06_{1.9}}$ | $50.8_{1.94}$ |
| − Transition | $95.24_{0.39}$ | $\mathbf{84.79_{0.86}}$ | $\mathbf{74.62_{1.27}}$ | $\mathbf{67.68_{1.82}}$ | $\mathbf{59.72_{4.1}}$ | $52.33_{4.45}$ |
| TLB | $76.66_{0.77}$ | $69.83_{1.36}$ | $66.22_{1.25}$ | $60.09_{0.23}$ | $55.36_{1.28}$ | $49.75_{1.65}$ |
| GUTLB | $76.17_{2.72}$ | $71.39_{1.65}$ | $67.37_{1.5}$ | $60.87_{1.6}$ | $\mathbf{57.95_{0.72}}$ | $52.36_{2.43}$ |

Table 2: Test accuracy of models trained on samples with $\leq 6$ logical operators and tested on out-of-distribution samples (number of logical operators between 7-12). * indicates that the results are taken from Shen et al. (2019a). † indicates that the results are taken from Tan et al. (2023). We bold the best results per block separately. We show the mean of 3 different runs. We bold the best results per block and the any results that overlap with it in their standard deviation. Subscript represents standard deviation. As an example, $90_{1.0} = 90 \pm 1.0$.

**Results:** We focus mainly on Transformer-based models here with a standard attention mechanism. We show the results in Table 1. Our Transformer baseline with xPos positional encoding is quite strong - outperforming prior report. Our UT baseline is a bit worse that prior report, but note that prior models had the advantage of being trained in the full training set without filtering higher length examples. Our proposed GUT performs the best in the near-IID setting, but like most other models, it is still poor in out-of-distribution generalization. Our ablations show each of our proposed modifications is helpful for GUT. Replacing a global halt with a token-level halt in GUT (GUT−Global Halt), removing the gating (GUT−Gate), or removing the transition dynamics information (GUT−Transition) - all harm the performance severely in the near-IID setting, and none improves the OOD performance substantially. The transition-based halting choice has the least impact on performance from the ablation.

Interestingly, TLB and GUTLB perform worse in the near-IID setting but better than others for OOD generalization. Their better generalization supports our intuitions that we discussed in §5.3 that chunk-wise recurrent with bound attention could be more robust. One reason why GUT and UT perform better in the near-IID settings could be that they better satisfy condition C1, which is beneficial for the structure of the task, but higher noise from dense softmax in longer contexts may still harm its length generalization (as we discussed in §5.3). Moreover, the chunking in TLB/GUTLB is not input-dependent - thus, may lead to bad chunks unsuited for the structure of the inputs. GUTLB does not benefit much over TLB.

## 6.5 Logical Inference

**Description:** Logical Inference (Bowman et al., 2015) is another structure-sensitive task where Transformers struggle to generalize (Tran et al., 2018). In this task, the model has to classify the logical relationship between two given sequences in propositional logic formalism. Examples of the data (taken from Tran et al. (2018)) are given below. Here, $(\sqsupset, \sqsubset)$ are entailment relations, and # implies logical independence. The

$$( \text{d} ( \text{or f} ) ) \sqsupset ( \text{f} ( \text{and a} ) )$$
$$( \text{d} ( \text{and} ( \text{c} ( \text{or d} ) ) ) ) \# ( \text{not f} )$$
$$( \text{not} ( \text{d} ( \text{or} ( \text{f} ( \text{or c} ) ) ) ) ) \sqsubset ( \text{not} ( \text{c} ( \text{and} ( \text{not d} ) ) ) )$$

models are trained on data with $\leq 6$ logical operators and tested on data with a higher number of operators 8-12.

|  | Length 512 | | | Length 1024 | | |
|---|---|---|---|---|---|---|
|  | | $p_i$ | | | $p_i$ | |
| **Model** | 0.1 | 0.8 | 0.98 | 0.1 | 0.8 | 0.98 |
| | | | Median | | | |
| Transformer | **100** | **100** | 90 | **100** | **100** | 87 |
| UT | **100** | 92.8 | 59 | **100** | 92.7 | 59 |
| GUT | **100** | **100** | 82 | 93 | **100** | 80 |
| TLB | **100** | **100** | 94 | **100** | **100** | 95 |
| GUTLB | **100** | **100** | **95** | **100** | **100** | **96** |
| TLB FR | **100** | **100** | 93.97 | 99.99 | 99.6 | 92.71 |
| | | | Mean and Standard Deviation | | | |
| Transformer | **$100_0$** | **$100_0$** | $89.83_{0.78}$ | **$100_0$** | $99.91_{0.12}$ | $83.88_{4.12}$ |
| UT | **$100_0$** | $92.81_{0.13}$ | $59.1_{0.20}$ | **$100_0$** | $91.65_{1.54}$ | $58.44_{0.66}$ |
| GUT | $99.15_{1.19}$ | $91.34_{12.23}$ | $73.08_{13.10}$ | $99.24_{1.04}$ | $88.48_{11.67}$ | $70.91_{12.99}$ |
| TLB | **$100_0$** | **$100_0$** | **$93.45_{1.85}$** | **$100_0$** | **$100_0$** | **$95.70_{1.28}$** |
| GUTLB | $83.19_{23.73}$ | $99.97_{0.02}$ | $89.07_{9.63}$ | $83.42_{23.37}$ | $99.94_{0.05}$ | $90.09_{9.38}$ |
| TLB FR | **$100_0$** | **$100_0$** | **$94.37_{1.99}$** | $99.99_{0.01}$ | $99.08_{0.86}$ | $92.94_{1.79}$ |

Table 3: Mean and median of 3 runs of models trained on 512 sequence length and $p_i = 0.8$ for flip-flop language modeling (Liu et al., 2023a). We bold the best results per block and any results that overlap with it in their standard deviation. Subscript represents standard deviation. As an example, $90_{1.0} = 90 \pm 1.0$.

**Results:** We show the results in Table 2. Here, both of our Transformer and UT baselines perform significantly better than prior reports. Besides that, we find similar results here as in ListOps. One interesting exception is that TLB and GUTLB perform worse in length generalization than others. This may be because the length generalization demanded in logical inference is less challenging (involving fewer sequence lengths) that listOps. Somewhat consistent with ListOps, TLB/GUTLB underfits in settings close to IID compared to other models, but starts to catch up at higher lengths. TLB/GUTLB may "fail" more elegantly on even longer sequence lengths (the kind that occurs in ListOps). Moreover, transition dynamics turn out to be less useful for GUT in this task. Sparse Universal Transformer (SUT) (Tan et al., 2023) recently achieved strong performance in this task. SUT uses a sparse Mixture of Expert (MoE) mechanism alongside dynamic halting. Instead of repeating the same layer, with MoE, SUT can dynamically select layer-specific experts to work with. It allows better scaling of parameters by increasing the number of experts without increasing the hidden state dimensions and the computational cost per layer. This is an orthogonal augmentation that We keep for future exploration (see §9).

## 6.6 Flip-flop Language Modeling

**Description:** Recently Liu et al. (2023a) showed that Transformer-based language models can be brittle. They proposed a minimal synthetic benchmark in the form of Flipflop language modeling that can diagnose some aspect of their architectural brittleness. An example of a sample from this task is "`w 0 i 1 w 1 r 1`". The write instruction (`w`) tells the model to write the following number into the memory, whereas the read instruction (`r`) tells the model to read the latest written number and print it out in the next position. The number following the ignore instruction (`i`) is to be ignored - essentially, they serve as distractors. The task can become challenging for Transformers as its attention mechanism has to ignore the influence of the distractors and zero in on the number associated with the last write instruction, irrespective of the sequence length and the receptive field of attention. We generate the data for this task as follows: We set $p_i$ as the probability of generating the ignore (`i`) instruction in any appropriate context. The probability for generating read or write instruction is then set as $(1-p_i)/2$ similar to Liu et al. (2023a). We set $p_i = 0.8$ and sequence length = 512 for the training data; we then test on samples with different $p_i$ ($p_i = 0.1, 0.8, 0.98$) and samples with both the same sequence length (512) and higher sequence length (1024). We generate $160,000$ training samples and $10,000$ samples per test split. The validation set is a mixture of $1,000$ samples

| Model | ListOps | Text | Retrieval | Image | Pathfinder |
|---|---|---|---|---|---|
| Transformer* | 36.37 | 64.27 | 57.46 | 42.44 | 71.40 |
| Transformer† | 38.37 | 61.95 | 80.69 | 40.57 | 65.26 |
| TLB† | 37.05 | 81.88 | 76.91 | **57.51** | **79.06** |
| Our Implementations | | | | | |
| Transformer | $45.05_{1.3}$ | $68.01_{0.78}$ | $78.65_{0.85}$ | $40.09_{0.52}$ | $70.26_{1.3}$ |
| UT | $28.50_{12.4}$ | $67.33_{0.8}$ | $79.15_{0.34}$ | $40.60_{0.029}$ | $67.17_{1.6}$ |
| GUT | $17.80_{0.0}$ | $67.95_{0.06}$ | $68.84_{12.9}$ | $34.82_{1.5}$ | $60.81_{3.5}$ |
| TLB | $\mathbf{50.35_{0.07}}$ | $\mathbf{83.55_{0.15}}$ | $\mathbf{86.48_{0.05}}$ | $53.39_{0.85}$ | $69.18_{13.0}$ |
| GUTLB | $40.88_{0.74}$ | $82.25_{0.26}$ | $85.09_{0.83}$ | $53.22_{0.43}$ | $69.02_{0.29}$ |

Table 4: Accuracy on LRA (Tay et al., 2021). * represents that the results are copied from (Tay et al., 2021). ‡ represents that the results are copied from (Chen et al., 2021). † represents that the results are copied from (Didolkar et al., 2022). We bold the best results per block and any results that overlap with it in their standard deviation. For our models, we report the mean of 3 runs. Subscript represents standard deviation. As an example, $90_{1.0} = 90 \pm 1.0$.

from each distribution setting evaluated in testing. We show the accuracy of the models in predicting the last read value. We generated the data using the same parameters as used in Liu et al. (2023a) except we also generated higher length test splits for checking length generalization capacity.

**Results:** We show the results in Table 3. We show both the median and mean in the table because some results have high variance. GUT again performs better than UT especially in longer sequences, but interestingly, basic Transformers outperform both GUT and UT, suggesting that both the latter models are potentially struggling to properly learn the halting function. We find both TLB and GUTLB perform quite well compared to simpler baselines and even GUT - supporting our hypothesis (§5.3) that chunk-wise recurrence may help make Transformers more robust by bounding the explicit size of the receptive field to a fixed chunk size - making it largely invariant to the change of sequence length. Thus, these models can also be considered alternatives to attention-sharpening techniques (Liu et al., 2023a).

Interestingly, Gated Universal Models (GUT, GUTLB) shows a bit of instability in this experiment, where the model degenerates in some runs (thus, high variance). For those cases, the median may better reflect how most of the time a "good run" would look like for these models.

To investigate the importance of bounding the chunk size more deeply, we also set up another model variant - TLB with fixed recurrence or a fixed number of chunks (TLB Fixed Recurrence in Table 3). In this model, instead of fixing the chunk size, we fix the number of recurrent steps (or, equivalently, the number of chunks) to five. For this model, Algorithm 2 is be modified such that $k$ would be directly set as 5 and made independent of sequence length ($len(H_0)$) and chunk size ($g$).

In the standard TLB, when sequence length increases, the number of chunks (and thus, the number of required recurrent steps) increases when the chunk size remains constant. Thus, the Transformer block has to manage the same length chunk in length-generalization settings. On the other hand, in TLB Fixed Recurrence, when sequence length increases, the chunk size increases, whereas the number of chunks (equivalently, the number of recurrent steps) remains constant. Here, the Transformers blocks have to handle larger length chunks in length-generalization settings.

Supporting our hypothesis further, TLB with fixed recurrence performs worse than TLB in length generalization ($length1024$) settings. This highlights the importance of maintaining an invariant chunk size over and beyond simply having non-linear recurrence. That said, some form of smarter input content-dependent dynamic chunking mechanism could be still helpful. It is something to consider in future work.

| Upperbound Layers | GUT No Halt Upperbound | GUT Upperbound | GUT |
|:---:|:---:|:---:|:---:|
| 10 | 2:27 min | 3:10 min | 2:77 min |
| 20 | 4:40 min | 6:10 min | 4:53 min |
| 30 | 6:52 min | 8:59 min | 4:20 min |
| 40 | 9:04 min | 11:57 min | 5:04 min |

Table 5: Training runtime on ListOps for different model per epoch. For GUT we report the average of 5 epochs.

### 6.7 LRA

**Description:** Long Range Arena (LRA) (Tay et al., 2021) is a useful set of tasks to investigate the long-range processing capabilities of different models. The task suite includes the following tasks:

1. **ListOps:** Same format as standard ListOps (Nangia & Bowman, 2018) but with longer sequence lengths ($\sim 2000$). Tested on IID test sets (no length generalization test involved).

2. **Text:** A byte-level sentiment classification task on IMDB movie reviews (Maas et al., 2011).

3. **Retrieval:** A byte-level document-pair classification task based on ACL Anthology Network (AAN) dataset (Radev et al., 2009) o determine whether two given documents are similar or not.

4. **Image:** An image classification task based on CIFAR10 (Krizhevsky, 2009). The images are flattened into a 1D sequence.

5. **Pathfinder:** An image classification task about determining whether a path exists between two points or not (Linsley et al., 2018). The images are again flattened into a 1D sequence.

We investigate the standard LRA task suite, which has no test set to check length generalization or Out-of-distribution generalization capacities.

**Results:** We evaluate our key models in the Long Range Arena (LRA) (Tay et al., 2021) in Table 4. Our xPos-based Transformer and TLB baselines, even with the same hyperparameters as in prior work, greatly outperform their prior reported performance and also newer baselines (Chen et al., 2021) in natural language tasks. However, we do not find the same benefit on the vision tasks with any of our models. In fact, some models like TLB with xPos perform significantly worse[5]. GUT mostly keeps up with Transformer, but as expected, it does not perform as well here in a more realistic setting due to limited parameters[6]. Surprisingly, GUT failed to train well in LRA ListOps compared to other models and is generally outperformed by UT. Leaving the ListOps subtasks aside, it is possible that GUT is not particularly helpful in LRA over UT because the other tasks in LRA are still relatively simple and do not require non-linear recurrence and dynamic halting. Notably, our xPos-based TLB shows the strongest performance in natural language tasks among all models, while GUTLB performs worse with its further parameter sharing. GUTLB possibly performs better than GUT because it has more parameters due to two separate attention layers for chunk processing and a separate memory update attention layer with different parameters.

## 7 Dynamic Halt Runtime Analysis

In Table 5, we analyze the effects of dynamic halt mechanisms on training runtimes. We compare three models: - 1) Gated Universal Transformer (GUT) that we discussed before, 2) GUT Upperbound - which is exactly the same as GUT but without any hard stop (no break statement) when the halting threshold

---

[5]To be fair, TLB's median on Pathfinder is still $\sim 77$. Only, in one run, it failed to learn the task.

[6]Similar limitations for UT were shown in prior works Narang et al. (2021). Matching the parameters will require either increasing per layer compute by increasing the hidden state size or some additional mixture of expert mechanism (Tan et al., 2023), which we keep for future exploration.

($\alpha_{thresh}$) is reached, and 3) GUT No Halt Upperbound - which is again like GUT but lacks any halting function altogether including any computation of halting probability or layer marginalization (it just returns the final layer after going through a fixed number of upperbound layers).

We compare the different models under different upperbound layers (10, 20, 30, and 40). GUT Upperbound is functionally equivalent to GUT. They share the same halting mechanism with layer marginalization at the end. But GUT upperbound has some redundant computation (that does not affect the output) of layers until the final upperbound layer is reached even if the halting threshold is exceeded. Thus, GUT Upperbound reflects the worst-case scenario (where GUT fails to halt at all) for GUT in terms of the empirical runtime.

GUT No Halt Upperbound shows the result of simply running GUT through the full upperbound number of layers without any halting mechanism at all. Thus, in this model, all the halting-related computations (e.g., per layer computation of halting probability) are removed, making each layer faster than that of the full GUT. Thus, GUT No Halt Upperbound can sometimes be faster than GUT even if the former goes through more layers (since each layer of the former is cheaper to compute).

We show the empirical training runtime of one epoch on ListOps for different models and different upperbound layers in Table 5. However, since the layers used in GUT are adaptive and can vary in different epochs, we report the mean empirical training runtime for the first five epochs in the case of GUT.

From the results, unsurprisingly, we find GUT upperbound tends to have more runtime than GUT. However, if the learned halting layer numbers for GUT tend to be not too far from the upperbound layer number, the difference between the three models can be minimal in terms of the runtime. Moreover, because GUT No Halt Upperbound has cheaper layers, it is sometimes empirically faster than GUT, even if GUT may halt a bit earlier. However, we note that as the upper bound is increased to 30 or 40, the runtime of GUT starts to saturate around 5 minutes, whereas the runtime of other models keeps increasing proportionally with the increasing upperbound.

Thus, in cases where we do not have a good estimate for the upper bound a priori or when the ideal number of layers varies wildly (with some examples requiring a low number of layers and some requiring much higher), it can be more viable to run GUT with a very high upper bound rather than GUT No Halt Upperbound (GUT without any halting mechanism) or other model setups.

## 8 Conclusion

In this paper, we compare side-by-side two forms of recurrences adapted to Transformers - depth-wise recurrence (with Universal Transformer (UT) as a representative) and chunk-wise recurrence (with Transformer Latent Bottleneck (TLB) as a representative). Furthermore, we investigate new extensions to depth-wise and chunk-wise recurrence by creating Gated Universal Transformer (GUT) representing depth-wise recurrence and Gated Universal Latent Transformer Bottleneck (GUTLB) representing chunk-wise recurrence. Following are the main takeaways:

**Length Generalization:** All the Transformer-based model do show some degree of length generalization (e.g. in logical inference and Flip Flop tasks), but it is still very limited. Particularly, all of them struggle to generalize at higher degrees - for example from 100 sequence lengths to 1000 sequence lengths or more in hierarchical tasks like ListOps.

**Depth-wise Recurrence:** GUT with global dynamic halting and gating generally tend to perform better than token-level halting in UT in algorithmic tasks. We also corroborate the results of Csordás et al. (2022) that a gating mechanism is beneficial as well. However, UT/GUT underperform (compared to chunk-wise recurrent models) in tasks like FlipFlop where C1 is not as important and a more chain-like sequential processing is suited. In the context of FlipFlop, what is ordinarily an advantage of GUT/UT (compared to GUTLB/TLB) in having the full context at every turn, can become a disadvantage by adding more noise for ignoring distractors.

**Chunk-wise Recurrence:** Generally, we find that chunk-wise recurrent models still struggle with tasks that demand C1 (like ListOps/Logical Inference). This is not completely surprising because the temporal recurrence constraint, in chunk-wise recurrence, directly conflicts with the ability to operate in arbitrary

orders (C1). In theory, however, an RNN-style setup with proper memory management could functionally simulate C1 Shen et al. (2019b) to an extent, but empirically without stronger inductive biases like in Ordered Memory Shen et al. (2019b), chunk-wise recurrent Transformers appear to fall short. The FlipFlop task does not require C1 and is more amenable to a chain-like sequential processing. As such, chunk-wise recurrence works well.

**TLB vs GUTLB:** Among the chunk-wise recurrent models (TLB, GUTLB), our current experiments show no benefit in using GUTLB over TLB. In fact, TLB generally is a bit better tahn GUTLB. However, this should not mean that we are to completely abandon the idea of GUTLB. First, the strong practical motivations of dynamic halting still applies as discussed in §4.1. Second, there could be still some key missing factor to unlocking the potential of GUTLB. This could be, for example, mixture of expert mechanism (as discussed more in §9), large-scale pre-training, or some better forms of memory management to counteract the limitation related to C1. As such, GUTLB (or something like it) may be worth revisiting at some point.

Overall, both chunk-wise and depth-wise recurrence have their own trade-offs. Currently the chunk-wise recurrent models perform better in LRA and FlipFlop tasks of ignoring distractors. They also fail more gracefully at higher lengths in ListOps. On the other hand, depth-wise recurrent models perform better near the IID regions of hierarchical tasks like ListOps and logical inference (chunk-wise recurrent models underfit the training distribution in these cases), while failing in tasks like FlipFlop. Given such results, for future model proposals, among other things, we would recommend evaluation on both ListOps and FlipFlop to check if we can have a unified model that can excel at both or achieve a better trade-off.

## 9 Limitations and Future Work

In this paper, we intended to do a focused study, limiting our scope to consider a few variables - mainly recurrence style and dynamic halting. As discussed below, there are other variables, tasks, and avenues to consider for future study.

**More tasks:** We only have a few experiments on larger realistic tasks. This is because Universal Transformer-based models are harder to scale parameter-wise (as discussed below) and tend to underperform in more realistic tasks where more parameters are beneficial (Narang et al., 2021; Tay et al., 2023). This would limit any insight we can gain from such tasks when comparing the different models. A potential approach can be to use something like a Mixture of Experts (as suggested below) for better scaling. Still, we keep such architectural choices for future investigation because they are out of the scope of our core focus in the chapter, which is about the recurrence structure and dynamic halting.

**Mixture of Experts (MoE):** The limitation of UT/GUT is scaling their parameters. The only way to increase their parameters is to increase per-layer compute (for example, by increasing hidden state size). However, since that same layer is repeated, the overall computation cost is increased by multiple folds compared to adding a new layer in a vanilla Transformer. One way to address this issue is to keep a set of neural modules and dynamically select a fixed number of a sparse subset of them in every layer (Cases et al., 2019; Pfeiffer et al., 2023; Le & Venkatesh, 2020; Shen et al., 2023; Tan et al., 2023; Rahaman et al., 2021; Weiss et al., 2022; Csordás et al., 2024). This way, we can increase the parameters by increasing the number of modules while keeping the per-layer compute roughly the same. Such strategies can be further explored in future works in combination with GUT. Such models can also be better explored in more realistic tasks.

**Alternative Attention:** We mainly focused on vanilla attention efficiently implemented using FlashAttention2 (Dao, 2024). However, there are other variations to consider that are understudied in some of these diagnostic tasks - for example, geometric attention (Csordás et al., 2022), linear attention (Yang et al., 2023; Qin et al., 2023a; Sun et al., 2023; Katharopoulos et al., 2020; Wu et al., 2022a), and others (Zimerman & Wolf, 2023; Tay et al., 2022).

**Recursive Structures**: Besides chunk-wise recurrence, it is also possible to explore more general chunk-wise recursive tree-like structures - for example, within a recursion-in-recursion framework (Ray Chowdhury & Caragea, 2023b) or using the strategy proposed by Chi et al. (Chi et al., 2023). Augmenting Transformers with a stack-based memory for simulating recursive capabilities can also be another avenue to explore (DuSell & Chiang, 2024; Li et al., 2024).

**Linear RNNs:** Recently, linear RNN based models are also gaining traction (Gu & Dao, 2024; Gu et al., 2022b; Martin & Cundy, 2018; Peng et al., 2023; Gu et al., 2022a; Orvieto et al., 2023; Smith et al., 2023; Qin et al., 2023b; 2024; Beck et al., 2024). There is room for further investigating them in similar contexts with dynamic halting and also further exploring RNN-Transformer hybridization (Ma et al., 2023; Ren et al., 2023; Pilault et al., 2023; Mehta et al., 2023; De et al., 2024; Lieber et al., 2024). However, some recent works suggest that Linear RNNs and adjacent models can still be limited compared to non-linear RNNs for state-tracking purposes (Merrill et al., 2024).

**Deep Equilibrium Networks:** Another alternative approach to dynamic halting or dynamic computation could be to frame Transformers as a Deep Equilibrium Network (Bai et al., 2019), which can implicitly and adaptively increase recursive iterations based on the sample input.

**Large Language Models and Chain of Thoughts:** Large Language Models (LLMs) with Chain-of-Thought (COT) reasoning prompts (Wei et al., 2022; Nye et al., 2021) and its extensions (Huang & Chang, 2023) can potentially counteract some of the original problems with fixed-layer Transformers (Zhou et al., 2023; Feng et al., 2023). Essentially, the chain of thought (literally like a chain of thought expressed in text) can serve as intermediate computation results (Merrill & Sabharwal, 2024). LLMs can dynamically vary the size of COT as per the input prompt, thus adapting computation based on the input. In essence, this can be an alternative way to incorporate adaptive computation into Transformers without any adaptive layer increase. However, Transformers may still benefit from dynamically varying the number of internal layers for more flexibility and for better capacity to learn from training examples that lack explicit COT style reasoning or model tasks where the dynamic reasoning aspect is not typically explicitly fully verbalized in natural symbolic language.

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

# A  Hyperparameter Details

For UT, GUT, or GUTLB the full training objective is: $\mathcal{L} = \mathcal{L}_{\text{main}} + \beta \mathcal{L}_{\text{ACT}}$. Here, $\mathcal{L}_{\text{main}}$ is the main task loss, and $\beta$ is a scalar trade-off factor for weighing $\mathcal{L}_{\text{ACT}}$. For ListOps (original), we use the same hyperparameters as used in the LRA version of ListOps Tay et al. (2021). We also keep the same hyperparameters for UT, and GUT as Transformers where possible. We tune the $\beta$ for UT based on validation accuracy on ListOps in $[1, 0.1, 0.01, 0.001]$ and set the value as 0.1. We use the same $\beta$ in other tasks as well, except in Logical Inference, where we found it better to set $\beta = 0$. We set $\alpha_{thresh}$ to 0.999. We use the same hyperparameters for TLB in ListOps as used in LRA ListOps for TLB in the original repository[7]. For GUTLB we set $L = 5$ but otherwise keep the same hyperparameters as TLB. We generally set $d_{gff} = d_{ff}$ for gated models. We augment all the sequences with a special START and END token before and after the input sequence for all tasks except flipflop. Generally, we use mean pooling for all models for all tasks, except for GUT/GUTLB, where we use the representation in the END position. The decisions are made based on the accuracy of validation on ListOps.

For Logical Inference, we use a Siamese model setup similar to Ray Chowdhury & Caragea (2021). For all the models, we use a mini-batch size of 256, a learning rate of $7e - 4$, an AdamW optimizer with weight decay $1e - 1$, and linear warmup for 4000 steps. For TLB/GUTLB, we use a chunk size of 10 and 10 memory slots. For UT or GUT, we use an upperbound layer as 15. Other setting details are consistent with ListOps unless mentioned otherwise above.

We use the same hyperparameters for Flipflop language modeling as Liu et al. (2023a). For TLB/GUTLB, we set the chunk size and memory slots as 10. For this task, we train the models in a causal language modeling setup. During training, we only send loss signals to predict the read outputs. The mean-based global halting in GUTLB and GUT is appropriately constrained in this context using cumulative sum to prevent leakage of future data.

For LRA tasks, we use the same hyperparameters as in the original LRA repository for the corresponding models for the natural language tasks. For GUT, we use an upperbound layer as 15. We transfer the rest of the hyperparameters of Transformers to GUT, and of TLB to GUTLB. One exception is that we use a bigger chunk size (100) for the Retrieval task for training efficiency for TLB/GUTLB than originally used; otherwise, they are exorbitantly slower to train. We set $L = 2$ for GUTLB for Retrieval/Text tasks in LRA. For Pathfinder in LRA, we found it better to use the hyperparameters from an alternative work Chen et al. (2021) for Transformer, UT, and GUT.

Each experiment was done in a single Nvidia RTX A6000.

---

[7] https://github.com/google-research/long-range-arena

