# OpenReview forum: "Investigating Recurrent Transformers with Dynamic Halt"
_TMLR — Rejected by TMLR_

### Review · Reviewer_yDzd · 2024-10-20

**Summary Of Contributions:**

This paper reviews two general approaches to incorporating recurrence with
transformer architectures, compares them empirically on a suite of benchmark
tasks. Motivated by the prospect of improving a transformer&rsquo;s ability to process
structured inputs by gating parts of hidden representations throughout the
recurrence, the authors propose novel modifications of the Universal Transformer
(UT) and Temporal Latent Bottleneck (TLB) architectures, serving as
representatives of depthwise and chunkwise recurrence in transformer models.
Upon evaluating the resulting models across a variety of benchmarks, the authors
critically evaluate the utility of depthwise and chunkwise recurrence for
various forms of generalization, as well as utility and computational expense
involved in recurrence halting mechanisms.

**Audience:**

Yes

**Broader Impact Concerns:**

No broader impact concerns.

**Claims And Evidence:**

No

**Requested Changes:**

1.  In the introduction, it says &ldquo;several works have questioned whether the
    complete elimination of recurrence has been the right move&rdquo;. Firstly, this
    sounds a little too casual. But more importantly, I think this should be
    explained even just briefly, for posteriority at the very least.
2.  In the introduction, it says &ldquo;[unconstrained all-to-all] interactions cannot
    accommodate for fixed layer depth&rdquo;. This should be explained more. What does
    &ldquo;unconstrained&rdquo; mean here? What is &ldquo;all tokens to all tokens&rdquo;? What is &ldquo;fixed
    layer depth&rdquo;? Why is this claim true?
3.  For depth-wise and chunk-wise recurrence, it would be very helpful to have a
    diagram depicting/comparing these architectures. Especially given that this
    paper is a long submission, there is room for such a diagram.
4.  In section 3, it would help to give textual descriptions of some of the
    parameters. For instance, if I am understanding correctly, $n$ is the context
    length and $d$ is the embedding dimension.
5.  The notation in equations 1-4 is confusing and probably slightly wrong. For
    example, I suspect equation (2) should say $H_{l+1} =
       \mathrm{FeedForward}(A_{l+1})$. Moreover, how exactly does equation (3)
    generalize equation (1)? What happened to the layer subscripts? It&rsquo;s also not
    exactly clear how the outer sum works &#x2014; what is the shape of the output of
    MHA? I realize this is fairly standard stuff, but if the equation is given,
    it should be precise.
6.  What is $d_{ff}$ (below equation (4))?
7.  For sections 4.1 and 4.2, there is no need for subsections &#x2014; just merge
    4.1.1 with 4.1 and 4.2.1 with 4.2.
8.  The discussion about bounding the number of recurrent applications in section
    4.1.1 is confusing. You first say &ldquo;we need to set some upper bounds&rdquo;. Then
    you say &ldquo;In the case of UT, we can set an arbitrarily high upper bound&rdquo;.
    Next, &ldquo;running every input through an arbitrarily high number of layers would
    be impractical&rdquo;. So&#x2026; you need to bound the number of recurrent
    applications, which was the first claim.
9.  The last sentence of the first paragraph in 4.1.1 appears to be broken.
10. Your use of the term &ldquo;layer&rdquo; in the UT is confusing. It appears you use
    &ldquo;layer&rdquo; and &ldquo;timestep&rdquo; interchangeably (I think &ldquo;timestep&rdquo; makes more sense
    for this purpose). To be more precise, in earlier sections, you refer to the
    recurrent applications of the Transformer cell as layers, while in 4.1.1,
    layers refer specifically to the individual layers within the Transformer cell.
11. In equation (5), you appear to be reusing the weights $W_1, W_2, b_1, b_2$
    from the description of the Transformer cells. I&rsquo;m assuming this is actually
    an additional set of weights, so the notation should reflect that.
12. In Algorithm 2, in the `for` loop with `enumerate`, the index $\mathbf{t}$
    should really be $t$.
13. I think the pseudocode of Algorithms 1 and 2 would be portrayed more
    effectively with block diagrams (especially Algorithm 2). This might be
    difficult for Algorithm 1 due to the thresholding mechanism, but for
    Algorithm 2 it should be straightforward.
14. At the end of page 6, you claim &ldquo;in practice, for hierarchical tasks [&#x2026;],
    the model may need to only temporarily restrict updates&#x2026;&rdquo; &#x2014; do you have
    a citation for this, or can you make this more precise? Intuitively, I do
    not agree. In such a scenario, why can&rsquo;t the model learn to not halt and
    simultaneously act as an identity map in the timesteps where it&rsquo;s waiting on
    some other computation? This whole paragraph lists many heuristic issues,
    but are these real issues in practice? If so, there should be citations.
    Else, why not demonstrate these phenomena in this paper?
15. What is the mean computed over in Algorithm 3? Is it a mean over tokens?
16. In section 6.1, is says &ldquo;For positional encoding, we mainly used xPos&rdquo;. This
    is confusing. It sounds like you&rsquo;re saying you sometimes *didn&rsquo;t* use xPos,
    and in this case, what did you use and why? Alternatively, maybe this was
    meant to mean &ldquo;we used xPos&rdquo; or &ldquo;we used xPos unless otherwise mentioned&rdquo;.
    That said, if it&rsquo;s the latter, I do not see any mention of what you ever use
    as an alternative to xPos (or any declaration of not using positional
    encoding). This should be clarified.
17. The experiment and results in section 6.4 should be clarified in several
    respects. For instance,
    1.  In the beginning of section 6.4, you say you train the transformers on
        expressions of length $\leq 100$. But in Table 1, the length for the
        &ldquo;near-IID&rdquo; column says $\leq 1000$. Is this a typo? If not, I am confused
        about what this column represents.
    2.  In the results, you claim that your UT baseline outperforms prior reports
        &#x2014; how is this justified in table 1?
18. Generally, in the tables, the &ldquo;bold&rdquo; designations do not take into account
    the confidence intervals. Datapoints whose confidence intervals overlap the
    respective bolded datapoints should also be bolded.
19. Table 3 should have confidence intervals.
20. Table 5 is very oddly formatted. It would be much easier to read if the rows
    were labeled by &ldquo;Upperbound Layers: $n$&rdquo; followed by the resulting
    datapoints for that number $n$ of layers.
21. The authors&rsquo; Transformer and UT implementations tend to outperform the
    existing reports by a very substantial margin. I would like to see more
    concrete discussion about why this is the case. What is the difference
    between your implementations and the baseline ones? Is it just the use of
    xPos positional encodings?
22. I think the conclusion is lacking quite a bit of structure and clarity.
    Section 6 presented a large variety of results, many of which suggested
    different conclusions about [G]UT and [G]UTLB. In section 8, it would be
    very helpful to have a sequence of paragraphs, each with a header describing
    a conclusion from the data, and containing pointers to the datapoints that
    lead to that conclusion. After having read sections 6 and 7, I&rsquo;m fairly
    overwhelmed trying to figure out what the takeaways should be from the
    results. It is still not entirely clear to me if there is any consensus
    about whether depthwise or chunkwise recurrence is preferred (or at least
    settings where one is preferred over the other), especially with regard to
    the gated variants introduced in this paper.

**Strengths And Weaknesses:**

## Strengths

The paper appears to conduct a fairly extensive review of recurrence and dynamic
halting mechanisms in transformers, which is helpful. The motivation for
studying these mechanisms, as well as motivations for their proposed extensions,
are clearly stated and reasonable. The experiments are conducted over a diverse
range of diagnostic tasks which are relevant for the purpose of diagnosing the
shortcomings and difficulties highlighted in their motivation.


## Weaknesses

My most severe weakness of the paper is a lack of cohesive takeaways /
conclusions about the results. After having read the paper, I still do not have
a clear idea about when one should prefer depthwise or chunkwise recurrence, as
well as whether or not the proposed gated variants are actually improvements.
While such discussions are present in subsections of section 6, their
justifications are not fleshed out in detail, and appear to be fairly speculative.
This could just be an issue with how the text is structured (see requested
changes). I strongly believe the general takeaways should be stated much more
explicitly.

I also found the clarity of the text to be lacking in many aspects (again, see
requested changes for examples). For instance, many mathematical expressions
were imprecise and/or underspecified. Moreover, *especially* given that this is a
&ldquo;long submission&rdquo;, several discussions could have been greatly simplified with
figures, for instance to depict the various architectures that are discussed
throughout the text. Another source of imprecision is in the description of the
experiments conducted in section 6&#x2014;for instance, there is confusion about how
some of the test sets are generated, and and with regard to confidence intervals.

---

> ### Author Response · Authors · 2024-11-18
> **Response [1/2]**
>
> We thank the reviewer for their detailed feedback and suggestions.
> We have overhauled the conclusion summarizing the main takeaways. We have temporarily highlighted the main changes in purple in the paper.
>
> 1. We removed that particular line. The lines following it already elaborate and expand on that (currently removed) line.
> 1. We expanded the context in the paper (Section 1 Introduction) - see [1] in response [2/2].
> 1. We will add diagrams.
> 1. Yes. We added the textual descriptions (updated in Section 3).
> 1. Thank you for pointing out the issues (updated in Section 3) .
>     1. You are right. It should be $FeedForward(A_{l+1})$. We changed it.
>     1. Eqn. 3 is not a generalization of Eqn 2. We changed the wording to better convey our intent: “The typical implementation of the $\mathrm{Attention}(Q,K,V)$ layer can be formalized as …”
>     1. We changed Eqn 3 to  $Attention(Q,K,V) = \mathrm{MHA}(LN(Q),LN(K),LN(V)) + Q$ which may make the lack of subscripts less confusing. We use subscripts in Eqns 1-2 to show how the output ($H_l$) from the previous layer ($l$) is being transformed to be the output ($H_{l+1}$) of the current layer ($l+1$). We explicitly set layer-subscripted inputs to Q,K,V arguments in Eqn 1 for that. However, in Eqn 3, we intend to show the function form of the attention function that is common to all layers without any layer-specific input values set. Thus, Eqn 3 lacks subscripts. We added layer subscripts again to Eqn 4.
>     1. We added the shape of MHA.
> 1. We added description of $d_{ff}$ (updated in Section 3). It’s just the size of the intermediate hidden states in the feedforward layer. It is essentially a hyperparameter whose value we provide in the appendix.
> 1. We merged the sections.
> 1. We overhauled that part (updated in Section 4.1) - see [2] in response [2/2]
> 1. We fixed it. Please see above.
> 1. We do not use time-step and layer interchangeably. By time-step we mainly refer to positions (we use time-step and position interchangeably). So time-step t is just the position t in the sequence. We use layer in the standard sense of the term - as some neural building block separated by non-linearities. The sense of “layer” doesn’t change from Transformers to UT.
> 1. Fixed.
> 1. Fixed.
> 1. Will add diagrams.
> 1. Note that simply the FFN simulating an identity map would not be enough here because what we would need is a selective identity map based on context, not a uniform application of identity transformation across all positions. That can be much more tricky to learn without better inductive bias. We expanded on that section (section 4.3) more in the paper, alongside citation - see [3] in response [2/2]
> 1. We use standard mean pooling in NLP that is typically done along the temporal axis. We added a more explicit formalization of the operation in Eqn 14 and some discussion surrounding it.
> 1. We removed “mainly”. xPos was used for all the models that we experimented with.
> 1. Clarifications:
>     1. This is not a typo. Training is indeed done with <= 100 sequence lengths. The near-IID split is simply the unfiltered original test split of ListOps that we keep for comparability with other works. The original test split has a mixture of multiple sequence lengths but higher sequence lengths are more rare. We say <= 1000, because that’s the upper bound. This is also the reason why we call it “near-IID” instead of IID simpliciter.
>     1. Thank you for catching this issue. The claim should have been meant for Logical Inference (Table 2). We adjusted that portion (section 6.4). “Our Transformer baseline with xPos positional encoding is quite strong - outperforming prior report. Our UT baseline is a bit worse that prior report, but note that prior models had the advantage of being trained in the full training set without filtering higher length examples.”
> 1. We will fix those.
> 1. Note that Table 3 was reporting median because some models had high variance. We, nevertheless, added mean + std performance alongside some added discussion.
> 1. Fixed.
> 1. In case of LRA, xPos should be the main reasoning since we otherwise use the same hyperparameters. In case of logical inference, the existing reports are probably based on poor hyperparameters (which were not shared).
> 1. We have overhauled the conclusion (section 8 conclusion).

---

> ### Author Response · Authors · 2024-11-18
> **Response [2/2]**
>
> [1] “As a motivating example, consider a task like ListOps (Nangia et al. 2018) where we have to solve a nested list of mathematical operations such as:
> MAX(1,3,SUM(4,5,MIN(9,7)),4)). Transformers utilize a self-attention mechanism where each position in a given sequence attends to all positions in the sequence. However, such an unconstrained all-to-all attention-based interaction is not immediately sufficient for a task like ListOps exemplified above. In the example, for instance, \verb+5+ cannot be summed up with MIN(9,7) immediately because the model has to wait until the MIN operation is applied. As such, the Transformer would need more layers to prepare intermediate computations (such as the output of MIN(9,7) before other outer operations can be applied and the final result is computed. Each position having access to all positions via self-attention is not enough to bypass this requirement. However, the number of sequential operations required to get the final result can vary from example to example whereas standard Transformers are typically bound by a fixed number of layers that does not depend on the input.”
>
> [2]  “However, in practice, we still need to set some upper bound for the number of layers because we do not want our models to add layers indefinitely. In the case of vanilla Transformers, we have to also worry about the number of parameters when setting the upper bound because more layers imply more parameters. In case of Universal Transformers, we do not need to worry about parameters to set the upper bound because the parameters are shared across all the layers. Nevertheless, we still need to worry about latency and computational resource when increasing layers. The standard strategy, given an upper bound, is to run through all the layers until the upper bound is reached. This strategy however is not ideal.  In practice, some inputs may require very little layers whereas some other inputs may require the maximum allowed number of layers. As such, the standard strategy results in a dilemma. Either, we can lower the upper bound to save compute and latency at the cost of ability to handle some of the more complex inputs or we have to keep a high upper bound to maintain coverage of more complex inputs but at the severe cost of latency and unnecessary compute for simpler inputs. Dynamic halting is a mechanism that provides a way out of this dilemma. The dynamic halting mechanism is designed to learn when to halt (i.e. learn in which layer to terminate) based on the given input. In the ideal setup, we can thus have a reasonably high upperbound to get coverage of complex examples, but at the same time, using the dynamic halting mechanism a model can  save on compute and latency by halting early in case of simpler examples.”
>
> [3] “However, in practice, for hierarchical tasks like mathematical reasoning or program synthesis, the model may need to only temporarily restrict updates at some positions (e.g., while waiting for computation in the inner lists to be completed in a task like ListOps). For example, given a sequence "4 x (5 + 6) - 7" (assuming a whitespace-based tokenization) states related to 4,x,-, and 7 has to wait until the first priority operation "(5+6)" is completed. While, in theory, Transformers may be able to implicitly learn a gating mechanism to preserve relevant information in some section of the hidden states across multiple layers, in practice having a more explicit mechanism can provide a better inductive bias in learning these desired capabilities. The benefit of an explicit gating mechanism for algorithmic tasks was empirically shown by  Csordas et al. 2022. “ The citation refers to: https://openreview.net/pdf?id=KBQP4A_J1K

---

### Review · Reviewer_tDxJ · 2024-10-30

**Summary Of Contributions:**

This paper reviews two types of methods to introduce recurrent mechanisms into transformers: (1) incorporating depth-wise recurrence (e.g., Universal Transformer), and (2) incorporating chunk-wise temporal recurrence (e.g., Temporal Latent Bottleneck). Then, the paper tries to extend and combine these methods, and investigates how each of these techniques influences the performance of the transformer on a series of synthetic tasks.

**Audience:**

Yes

**Claims And Evidence:**

Yes

**Requested Changes:**

1.	(See weakness 1 for details) Clarify which part is addressing the claim on inductive biases, and add a piece of conclusion for each inductive bias as takeaway of the paper. Reorganize the experiments if possible.

2.	(See weakness 2 for details) Give fine-grained analysis on some specific examples.

3.	The motivation of using xPos as the positional encoding is not well explained. We can see from Table 1-4 that with the use of xPos, the performance of the standard transformer and UT significantly improved over their original implementation. It makes me wonder (1) why this change of positional encoding is necessary for the experiments, because the paper mentioned in the last line of Introduction that the main purpose of this paper is not to chase the state-of-the-art, (2) how this change of positional encoding influence the performance and “inductive bias” of the new methods proposed in this paper (GUT and GUTLB). I encourage a comparative study on different positional encoding approaches to see if the current conclusions still hold, or a detailed discussion on the motivation of using xPos as the positional encoding.

4.	Could you explain why in Table 3, results on a small $p$ ($p=0.1$) is usually better than results on an extremely large $p$ ($p=0.98$). Since the model is trained with $p=0.8$, it seems to imply that transformers trained on sequences with a moderate distraction ratio $p=0.8$ is capable of generalizing to sequences with a small distraction ratio $p=0.8$, but not to sequences with a small distraction ratio $p=0.98$.

5.	Could you explain why in Table 5, the training runtime of GUT decreases when the number of upperbound layers is increased from 20 to 30?

6.	Minor: issues with notations.

a)	Inconsistent use of bold symbols. In Algorithm 1 and Equation (7), both bold symbols $\boldsymbol{s}_l^{(t)}, \boldsymbol{h}_l^{(t)}$ and light symbols ${s}_l^{(t)}, {h}_l^{(t)}$ are used. If they have different meanings, please clarify this. If they have the same meaning, please consider use the same symbols.

b)	Are the $W_1$ and $W_2$ in Equation (5) the same as the $W_1$ and $W_2$ in Equation (4)? If not, please consider using different	 notations.

**Strengths And Weaknesses:**

Strengths:

1.	The structure of the whole paper is clear and makes it quite easy to follow.

2.	The background and preliminary parts are detailed enough for readers who are not familiar with this field.

Weaknesses:

1.	I’m struggled to identify which part of the paper is addressing the claim “we comprehensively study the inductive biases of ….”

a)	I first thought Section 4.3 which discussed the limitations of existing methods (UT and TLB) is the part that explains the “inductive biases.” However, it is not clear whether Section 4.3 is truly the contribution of this paper or just commonly recognized facts in previous literature.

b)	Then, I thought the case studies in Section 6 is the part that explains the “inductive biases.” However, it is still not clear “what” kind of inductive biases each method exhibits. There are no clear conclusions and takeaways such as “SGD prefers flatter local minima.”

c)	I suggest the authors make it clear which part is addressing the claim on inductive biases, and add a piece of conclusion for each inductive bias as takeaway of the paper. I also encourage a reorganization of the experiments: organizing by insights/conclusions rather than organizing by datasets, i.e., each subsection first presents an insight/conclusion, then provide empirical support for this insight/conclusion on various datasets.

2.	The experiments mainly focus on the *performance (accuracy)* of different methods on four synthetic tasks. However, I think the performance alone is not enough to investigate the inductive bias and the detailed inference logic of a model. It is better to do some fine-grained analysis on several examples to show that the model truly learns something to adapt to the input structure. For example, in the task of ListOps, it might be interesting to check for a nested list of depth 10, how many layers does a UT run on each token (is it also 10 layers?), and how this result changes for a GUT with gating and global halting. This kind of illustrating examples could offer a more in-depth and intuitive understanding of what’s happening in the transformer model.

---

> ### Author Response · Authors · 2024-11-18
>
> We thank the reviewer for their constructive suggestions. We respond to the concerns raised by the Reviewer below. We updated the draft accordingly. We have temporarily highlighted the main changes in purple.
>
>
> 1. Points:
>     1. We rephrased it to “In this paper, we study the empirical effects of two major approaches to augmenting Transformers with a recurrent mechanism“ and removed the mention of “inductive biases” from other places to reflect our aim more directly (Abstract, Section 1 Introduction)..
>     1. We overhauled the conclusion with clearer takeaways (Section 8 Conclusion).
> 1. We will consider adding more analysis.
> 1. We explore xPos in the paper because it is a well-received popular positional encoding that is specifically motivated for length extrapolation - which is the regime we explore. You are right that our goal is not necessarily to have the strongest SOTA setup with all of the best available extensions, but we still wanted to keep reasonably strong baselines. Our aim was to fix a reasonable positional encoding (chosen as xPos) and explore other variables that are of interest for our scope of focus. In terms of comparison, Table 4 shows some results that illustrate the effect of xPos on Transformers. Particularly, compare our Transformers (with xPos) under “Our Implementations” and Transformers from a prior report (marked by the $\dagger$ sign - second row in the first block; this is without xPos but with similar hyperparameters).
> 1. Your intuition is correct. But as we define it (“We set $p_i$ as the probability of generating the ignore (\verb_i_) instruction in any appropriate context.”), p represents the probability of distractors (i instructions). So p=0.98 is the setting with the highest density of distractors - thus, the lowest performance consistent with your intuition.
> 1. Since the learning of dynamic halting is also conditioned on stochastic factors like random initialization, even with higher upper bound the model can learn to halt slightly earlier on average (due to stochastic deviations) - slightly reducing the training runtime.
> 1. Notations:
>     1. We normalized the notations. Bold vs non-bold are the same.
>     1. They are different. We fixed the notation.
>
> Minor correction: note that not all the tasks are synthetic. For example, in LRA, the Text task is based on IMDB text classification (natural language movie texts). And AAN is a task of sentence similarity classification involving natural language documents. IMG is based on CIFAR10 image classification task - which is also non-synthetic.

---

> > ### Comment · Reviewer_tDxJ · 2024-11-23
> >
> > Thank you for the response. I have no further questions. I will take the revisions into account when making the recommendation.

---

### Review · Reviewer_bYmp · 2024-11-02

**Summary Of Contributions:**

This paper compares the capabilities and inductive biases of modified
transformer architectures which are extended to include recurrence in
their evaluation. The authors study two general approaches (depth-wise
and chunk-wise recurrence) as well as a combination of these two and
modified halting mechanisms. The authors contextualize these by
discussing some of the desired capabilities which may be beneficial
for complicated tasks and compare performance on a collection of
benchmark problems.

**Audience:**

Yes

**Broader Impact Concerns:**

No concerns

**Claims And Evidence:**

Yes

**Requested Changes:**

I appreciate the clear exposition and presentation of your
architectural changes. Below are a few questions and notes that I
think could enhance clarity.

*(Important)* I think the most important improvement may be to enhance
the clarity of your conclusions and any takeaway points. In
particular, is there any guidance that a reader could derive from your
results to guide their own work?

You have a good collection of results and some commentary on what
factors may have caused some of your proposed architectures to perform
better/worse on particular problems, but are there general trends you
can observe from this (for example for other tasks)? I think making
this clearer would be helpful especially since you note that your goal
is to study the inductive biases of these transformer augmentations
and you also note at the end of section 1 that the GUTLB extensions do
not seem to provide a significant advantage.

I think some of this material is present (although perhaps not evenly
across all experiments), but collecting it or making it clearer could
help strengthen the conclusions from your results.

In particular, I appreciate the extended discussion for your results
on ListOps in section 6.4 and the flip-flop modeling in 6.6. However,
it seems to me that discussions of logical inference in 6.5 are maybe
a bit less developed. Are there trends there that could be further
discussed?

*(Moderate)* It might also help readers to include a (possibly /very/
brief) introduction to the tasks. Currently some of these aren't
terribly self-contained. Your descriptions of ListOps and flip-flop
modeling I think are clear, but some of the others might benefit from
being extended a bit (LRA and logical inference). If the descriptions
are hard to make sufficiently short I think some of this could be
pushed to an appendix section.

*(Minor)* when discussing $\mathcal{L}_{\text{ACT}}$ it may help to
note that this requires tuning. I think it's ok to leave the $\beta$
parameter tuning details to the appendix, but it may help to note that
it is present to give a full picture of the loss used during training.

*(Minor)* The note in section 6.5 about "recently achieved strong
performance... but uses a sparse MoE" I think needs a bit more
discussion if it is kept at this point in the document. I don't
believe "MoE" is mentioned elsewhere until the future work section
after the conclusion.

*(Minor)* It may be worth checking the configuration of the headings in
the appendix. At the moment it appears that appendix section A is
empty and the only content is in appendix section B.

*(Question)* In section 7 is there a reason to use the training
runtimes, vs some measurement of inference time?

**Strengths And Weaknesses:**

**Strengths**

- Clear exposition and good motivation for extensions
- Good discussion of some of the factors that may be contributing to
  the performance of the various architectures on the benchmark tasks
- Variety of benchmark problems for testing extensions

**Weaknesses**

- Conclusions and takeaways could be more clearly distilled
- Explanation of the tasks could be more self-contained

---

> ### Author Response · Authors · 2024-11-18
>
> We thank the reviewer for their appreciation of the paper and useful feedback.
> We have temporarily highlighted the main changes in purple.
>
> Our updates based on your suggestions:
>
> 1. **Better conclusion:** We have overhauled the conclusion (Section 8 Conclusion)
> 1. **Task Descriptions:** We have added more description for Logical Inference (Section 6.5 Logical Inference) and LRA tasks (Section 6.7 LRA).
> 1. **More discussions on Logical Inference Experiments:** We added more discussions (Section 6.5 Logical Inference) . However, note that most of the trends in listops spill-over for logical inference so it is relatively short.
>
> **Q. Why training time latency (not inference):** This was not a principled choice. It just something that immediately came to our mind and was convenient to set up.
>
> We also addressed all the other minor points.

---

> > ### Comment · Reviewer_bYmp · 2024-11-25
> >
> > Thank you for your response and your work on the revisions. I
> > appreciate your efforts to expand the paper and address comments.
> >
> > I have some follow-on concerns relating to the training time
> > measurements. I guess my remaining suggestion is that it should
> > perhaps be made clearer what conclusions should be drawn from your
> > measurements and how the training inference timing supports that (vs. perhaps
> > taking measurements after training). I think there are also possible
> > concerns relating to *which* training epoch was used for measurement (in
> > particular was it early or late in the training process?) and how
> > averaging--it seems--over multiple training steps might complicate
> > analysis (i.e. I imagine the network weights are changing between some
> > of these timing measurements?).
> >
> > Thanks again for your work during revision.

---

### Decision · Action_Editor_oUTP · 2024-12-09

**Recommendation:** Reject

**Comment:**

This paper presents a comparison of two different ways of introducing a recurrent mechanism into transformer architectures (Universal Transformer and Temporal Latent Bottleneck) and then proposes some new variations and extensions.  The paper seems well motivated and the topic would be interesting to the community given the popularity of transformer architectures.  The reviewers liked the review of recurrent architectures and dynamic halting, and that the authors evaluated a diverse set of tasks.  The reviewers all found that the paper would be relevant to the community and two of the three found that the claims justified the evidence.  However, the reviewers all voted to reject the paper (Leaning Reject, Reject, Leaning Reject).  It seems that the main issues come from an overall lack of clarity and a lack of clear and well-justified takeaways.  There were some concerns about the presented experiments and whether the metrics were well justified (e.g. training time vs inference time as a metric).  One reviewer noted many issues with mathematical notation and technical presentation.

The presented topic is clearly of interest to the community, but it seems like this paper falls short of the potential impact of this study.  The reviewers all struggled to understand what the takeaways were.  This seems to come from an overall lack of clarity in the paper, but also somewhat speculative conclusions.  The reviews and discussions give a sense that the paper is incomplete.  It requires some more revisions for clarity, some more analysis of the results and a more clear exposition of the takeaway messages.  The paper also doesn't have any figures, which might be part of the reason it's hard to follow?  Figures could go a long way towards explaining the empirical results and takeaway messages more clearly.   The required changes are beyond minor revisions.  Therefore, I would recommend rejecting the paper.  If the reviewers adequately addressed the reviewers concerns, I think this could be a strong re-submission at a future date.

**Audience:**

Transformers are the dominant architecture in the massive wave of large foundation models, so this is clearly relevant to the community.   However, the reviewers all noted that they didn't really understand the takeaways from the paper.  Clearly, that limits the potential influence this would have on the community.  It seems like the topic is very relevant and hot, but due to clarity issues this paper might fall flat.   That alone seems like a reason to do some significant rewriting.

**Claims And Evidence:**

Two out of three of the reviewers found that the claims were justified by the presented evidence.  However, in reading the reviews it doesn't seem like the reviewers were entirely convinced.  The reviewers seemed to struggle to understand the takeaways from the paper, there were no figures to provide clarity, and they seemed to disagree with some of the ways the authors measured differences between architectures (e.g. training time).

**Resubmission Of Major Revision:**

The authors may consider submitting a major revision at a later time.